# Distinct Pathological Changes in Preweaning Mice Infected with Live-Attenuated Rift Valley Fever Virus Strains

**DOI:** 10.3390/v16070999

**Published:** 2024-06-21

**Authors:** Cigdem Alkan, Eduardo Jurado-Cobena, Tetsuro Ikegami

**Affiliations:** 1Department of Pathology, The University of Texas Medical Branch at Galveston, Galveston, TX 77555, USA; cialkany@utmb.edu; 2Department of Microbiology and Immunology, The University of Texas Medical Branch at Galveston, Galveston, TX 77555, USA; ejcobena@utmb.edu; 3Center for Biodefense and Emerging Infectious Diseases, The University of Texas Medical Branch at Galveston, Galveston, TX 77555, USA; 4Sealy Institute for Vaccine Sciences, The University of Texas Medical Branch at Galveston, Galveston, TX 77555, USA

**Keywords:** Rift Valley fever, live-attenuated vaccine, MP-12, RVax-1, ∆NSs-∆NSm-rZH501, mouse, encephalitis, hepatitis, preweaning, age dependent

## Abstract

Rift Valley fever (RVF) is a mosquito-borne zoonotic viral disease endemic to Africa and the Middle East. Live-attenuated RVF vaccines have been studied for both veterinary and human use due to their strong immunogenicity and cost-effective manufacturing. The live-attenuated MP-12 vaccine has been conditionally approved for veterinary use in the U.S.A., and next-generation live-attenuated RVF vaccine candidates are being actively researched. Assessing the virulence phenotype of vaccine seeds or lots is crucial for managing vaccine safety. Previously, preweaning 19-day-old outbred CD1 mice have been used to evaluate the MP-12 strain. This study aimed to characterize the relative virulence of three live-attenuated RVF vaccine strains in 19-day-old inbred C57BL/6 mice: the recombinant MP-12 (rMP-12), the RVax-1, and the ∆NSs-∆NSm-rZH501 strains. Although this mouse model did not show dose-dependent pathogenesis, mice that succumbed to the infection exhibited distinct brain pathology. Mice infected with ∆NSs-∆NSm-rZH501 showed an infiltration of inflammatory cells associated with infected neurons, and focal lesions formed around virus-infected cells. In contrast, mice infected with rMP-12 or RVax-1 showed a minimal association of inflammatory cells in the brain, yet the virus spread diffusely. The preweaning model is likely useful for evaluating host responses to attenuated RVFV strains, although further refinement may be necessary to quantitate the virulence among different RVFV strains or vaccine lots.

## 1. Introduction

Rift Valley fever (RVF) is a zoonotic viral disease primarily found in sub-Saharan Africa, Egypt, the Comoros, Saudi Arabia, and Yemen [1]. The causative agent, the RVF virus (RVFV), belongs to the *Phlebovirus* genus within the Phenuiviridae family. The RVFV genome is composed of three RNA segments: large (L), medium (M), and small (S). The L-segment encodes the viral RNA-dependent RNA polymerase, while the M-segment encodes the envelope proteins Gn and Gc, the 78 kD protein, and the non-structural M (NSm) protein [2,3]. The S-segment encodes the nucleoprotein (N) and the nonstructural S (NSs) protein.

Livestock, including sheep, cattle, goats, and camels, are highly susceptible to RVF, exhibiting symptoms such as abortions, fetal demise, fetal malformations, and a significant lethality rate among newborn animals. Humans can contract the RVFV through mosquito bites or contact with bodily fluids from infected animals. Human RVF typically manifests as a self-limiting febrile illness, though some cases progress to hemorrhagic fever, encephalitis, or retinitis [4]. Meanwhile, floodwater *Aedes* spp. mosquitoes can vertically transmit the RVFV, with infected eggs capable of surviving in dry environments for extended periods. Currently, there are no licensed RVF vaccines for humans, though a few are approved for use in certain endemic regions [5,6]. The live-attenuated Smithburn vaccine, derived from the mouse neuro-adapted Entebbe strain, has been utilized for livestock animals in endemic countries for decades. However, concerns regarding its significant residual virulence in pregnant animals have been raised. More recently, the live-attenuated Clone 13 vaccine, featuring a 69% in-frame deletion of the NSs gene in the 74HB59 strain, was approved for use in livestock animals in South Africa and other regions. Nevertheless, implementing a routine RVF vaccine is challenging due to the infrequent occurrence of outbreaks, which often results in delays in vaccination efforts when an outbreak does occur.

Due to its substantial public health and agricultural implications, the RVFV is classified as a Category A Priority Pathogen by the National Institute of Allergy and Infectious Diseases (NIAID) and designated as an overlap select agent by the U.S. Department of Health and Human Services (HHS) and Agriculture (USDA) in the United States. RVF is also listed as a priority disease in the World Health Organization’s (WHO) R&D Blueprint and is a notifiable disease to the World Organization for Animal Health (WOAH). RVF vaccines for humans and animals are crucial countermeasures against the potential introduction of the RVFV into non-endemic countries. In the U.S., the live-attenuated MP-12 vaccine, derived from the mutagenized ZH548 strain [7], has been licensed conditionally for livestock use [8]. The single dose vaccination with MP-12 could induce rapid and long-term protective immunity in livestock animals and humans [9,10]. The MP-12 strain encodes 9 amino acid substitutions, 10 synonymous mutations, and 4 mutations in untranslated regions. The attenuation of the MP-12 strain is largely attributed to a few amino acid changes in the M- and L-segments, while the S-segment is less contributing to the attenuation: i.e., Y259H (Gn), R1182G (Gc), and R1029K (L) [11]. Additionally, two mutations within the L-segment (V172A and M1244I) contribute to the temperature-sensitive phenotype of the MP-12 strain, restricting virus replication when temperatures exceed 38 °C [12].

The potential drawbacks of the veterinary MP-12 vaccine are (i) the lack of serological markers to differentiate infected animals from vaccinated ones (DIVA) and (ii) genomic RNA cannot be distinguished from the pathogenic RVFV by PCR screening. To address these issues, the next-generation MP-12 vaccine, RVax-1, has been developed. This strain features 36, 37, 167, and 326 silent mutations in the open reading frames of the N, NSs, M, and L genes, respectively, which serve as genomic markers to distinguish it from wildtype RVFV strains. Additionally, these modifications enhance its attenuation phenotype through changes in the MP-12 S-, M-, and L-segments [13,14]. Furthermore, RVax-1 incorporates the deletion of the 78 kD/NSm gene, which serves as a serological DIVA marker and reduces the dissemination of the vaccine virus in mosquito vectors. RVax-1 retains the NSs gene because deleting the NSs gene from the MP-12 backbone results in a reduced immunogenicity of the vaccine. Meanwhile, the recombinant ZH501 strain with deletions in the NSs and 78 kD/NSm genes (∆NSs-∆NSm-rZH501) has shown significant attenuation in livestock animals and is considered a promising RVF vaccine candidate for both animal and human use [15,16]. The RVFV NSs protein is a key virulence factor, aiding viral evasion from host innate immune responses by degrading TFIIH p62 protein and PKR through proteasomal mechanisms [17,18,19,20,21,22]. Consequently, the ∆NSs-∆NSm-rZH501 exhibits significant attenuation in immunocompetent hosts. Additionally, the absence of the 78 kD/NSm gene reduces the dissemination of the vaccine virus in mosquito vectors [3,23].

Regardless of age, mice display a notable susceptibility to the wildtype RVFV infection. Even a minimal number of plaque-forming unit (PFU) can lead to a lethal disease marked by acute fulminant hepatitis and delayed viral encephalitis. Conversely, the live-attenuated MP-12 strain demonstrates significant attenuation in adult mice, with LD_100_ values of 1 PFU for ZH501 and LD_50_ values of ≥10^6.3^ PFU for MP-12 [7,24]. Notably, the susceptibility of mice decreases in an age-dependent manner, with younger animals being more prone to infection. Prior research indicates that preweaning 19-day-old outbred mice exhibit moderate susceptibility to the MP-12 strain, allowing for the measurement of relative virulence among its variants: i.e., LD_50_ of MP-12 = 75.9 PFU in 19-day-old CD1 mice [10]. In this study, we characterized the susceptibility of 19-day-old preweaning inbred C57BL/6 mice, instead of outbred CD1 mice, to maintain a uniform genetic background for comparison among groups. We evaluated three live-attenuated RVFV strains: the recombinant MP-12 strain (rMP-12), the RVax-1 strain, and the ∆NSs-∆NSm-rZH501 strain. While the preweaning inbred mice exhibited high susceptibility to all three strains, noticeable pathological differences were observed among the infected animals.

## 2. Materials and Methods

### 2.1. Media, Cells, and Viruses

BHK cells that stably express T7 RNA polymerase (BHK/T7-9 cells) [25] were maintained in MEM alpha containing 10% FBS, penicillin (100 U/mL), streptomycin (100 μg/mL), and hygromycin B (600 μg/mL). Vero cells (ATCC CCL-81) were maintained at 37 °C in DMEM (Gibco, Thermo Fisher Scientific Inc., Waltham, MA, USA), containing 10% fetal bovine serum, penicillin (100 U/mL, Gibco), and streptomycin (100 µg/mL, Gibco), in a humidified cell culture incubator with 5% CO_2_. The recombinant RVFV MP-12 vaccine strain (rMP-12) and RVax-1 strain were recovered from Vero cells via reverse genetics, as described previously [13,26]. The recombinant ZH501 strain lacking NSs, NSm, and 78 kD genes (∆NSs-∆NSm-rZH501) was successfully generated from BHK/T7-9 cells. This was achieved through co-transfection with various plasmids: pProT7-wS(+)∆NSs, which encodes the deletion of the entire NSs region, identical to that of MP-12 lacking the entire NSs gene [27]; pProT7-wM(+)-M847A-∆NSm21/384 [28]; and pProT7-wL(+), pT7-IRES-wN, pT7-IRES-wL, and pCAGGS-wG [29]. The titration of RVFV was performed by a plaque assay using Vero cells with crystal violet or neutral red staining [30,31]. Cells and viruses used in this study were verified to be mycoplasma free at the University of Texas Medical Branch at Galveston (UTMB) Next Generation Sequencing Core Facility or Tissue Culture Core Facility.

### 2.2. Mouse Challenge Experiment Using Live-Attenuated Rift Valley Fever Viruses

Timed-pregnant C57BL/6NCrl mice (*n* = 10) were obtained from Charles River Laboratories. At the Animal Biosafety Level 2 (ABSL2) laboratory at the University of Texas Medical Branch, nine groups of preweaning mice at 19 days of age were challenged intraperitoneally (i.p.) with either rMP-12, RVax-1, or ∆NSs-∆NSm-rZH501 at the designated doses (Table 1). The virus titers in those inocula were re-evaluated through back titration via a plaque assay. Furthermore, mother mice were also subjected to the viral challenge. Health status and body weight of those mice were observed for 28 days post-challenge (dpc). Mice showing more than 20% body weight loss and/or showing clinical signs such as severe lethargy, hunched posture, paralysis, or reluctance to move in response to stimulation were humanely euthanized, while all surviving mice were humanely euthanized at 28 dpc. Tissues including liver and brain from animals were homogenized with 2.8 mm stainless steel beads (Millipore-Sigma, Burlington, MA, USA) in Trizol Reagent (Life Technologies, Carlsbad, CA, USA) or PBS. The homogenates were then analyzed to measure viral RNA load by reverse transcription quantitative polymerase chain reaction (RT-qPCR) or to determine infectious virus titers by a plaque assay, respectively. Post-mortem samples were fixed in 10% neutral buffered formalin for histopathological analysis.

### 2.3. RT-qPCR Measurement of Viral RNA Loads in Tissues

Total RNA was extracted from tissue homogenates in Trizol using the Direct-zol RNA Miniprep Kit (Zymo Research, Irvine, CA, USA). First-strand cDNA was synthesized from total RNA using iScript Reverse Transcriptase (Bio-Rad Laboratories, Hercules, CA, USA), followed by PCR amplification with SsoAdvanced Universal Probes Supermix (Bio-Rad Laboratories) on a Mic qPCR Cycler (4 channels), as previously described [13]. The PCR reaction targeted the 5′ untranslated region of M RNA using the forward primer (RV-MUTR-F), the reverse primer (RV-MUTR-R), and a Taqman probe [13]. A standard curve generated from cDNA sets derived from serial RNA dilutions with known concentrations was employed for validating the sample RNA copy numbers.

### 2.4. Histological Examination

Mouse tissues, including livers, spleens, and brains, which were fixed in 10% neutral buffered formalin, were embedded in paraffin blocks, sectioned, and stained with hematoxylin and eosin (H.E.) at the Anatomic Pathology Laboratory at UTMB. For immunohistochemistry (IHC), the sectioned tissues were treated with proteinase K antigen retrieval solution (Abcam, Waltham, MA, USA), followed by blocking with Animal-Free Blocker and Avidin/Biotin Blocking Kit (Vector Laboratories, Inc. Newark, CA, USA). Subsequently, they were incubated with either anti-RVFV N rabbit polyclonal antibody [32] or anti-mouse CD68 antibody (ab283654, Abcam), followed by a biotinylated secondary antibody (Vector Laboratories, Inc.). Sections were then treated with streptavidin alkaline phosphatase (Vector Laboratories, Inc.) and ImmPACT Vector Red substrate (Vector Laboratories). Finally, background staining was visualized by additional hematoxylin staining. Images were captured using Olympus cellSens software version 3.2 with a DP71 camera attached to an Olympus IX73 microscope. Color deconvolution of the captured TIFF images (liver sections stained with anti-CD68 antibody) was subsequently performed using FIJI ImageJ 1.53a software [33]. The red color channel was selected for analysis. The image threshold was adjusted to highlight the intensity of positive signals, and the converted black-and-white image was then used to measure the particles.

### 2.5. Statistical Analysis

The research data underwent statistical analysis using GraphPad PRISM software (version 8.4.3). The survival curves of mice were generated using the Kaplan-Meier method and subsequently analyzed using a log-rank test. Group comparisons were conducted employing either one-way ANOVA, followed by Tukey’s multiple comparison test or two-way ANOVA and then by Sidak’s multiple comparison test.

### 2.6. Ethics Statement

All experiments involving recombinant DNA and infectious RVFV were conducted with approval from the Institutional Biosafety Committee (IBC) at UTMB, as specified in Notification of Use (Approval #2021017). The use of the ∆NSs-∆NSm-rZH501 strain in a BSL2 laboratory by T.I. has received approval from both the NIH Office of Science Policy and the IBC at UTMB. Mouse studies were conducted in the UTMB Building 17 ABSL2 laboratory, accredited by the Association for Assessment and Accreditation of Laboratory Animal Care (AAALAC). Animal protocol #1912097 was approved by the UTMB Institutional Animal Care and Use Committee (IACUC).

## 3. Results

### 3.1. Susceptibility of 19-Day-Old C57BL/6 Mice to Live-Attenuated Rift Valley Fever Viruses

Nineteen-day-old C57BL/6 inbred mice, prior to weaning, were intraperitoneally challenged with live-attenuated RVFV strains, namely rMP-12, RVax-1, or ∆NSs-∆NSm-rZH501, to assess their relative virulence in this model, as detailed in Table 1. Mice were monitored for 28 days following virus challenge. As shown in Figure 1a,c,e, body weight gains were similar between the rMP-12 and the RVax-1 vaccinated groups, whereas the ∆NSs-∆NSm-rZH501 group showed less body weight than the rMP-12 or the RVax-1 groups. Survival curves were generated using the Kaplan–Meier method (Figure 1b,d,f), and the log-rank test was employed to assess statistical differences among the groups. There were, however, no statistically significant differences observed in the survival curves among the rMP-12, RVax-1, and ∆NSs-∆NSm-rZH501 groups. Mouse deaths were recorded between 5 and 16 dpc for rMP-12; 6 and 14 dpc for RVax-1; and 5 and 8 dpc for ∆NSs-∆NSm-rZH501. The survival rates were as follows: for rMP-12, 50% (1000 PFU), 33.3% (50 PFU), or 57.1% (10 PFU); for RVax-1, 42.9% (1000 PFU), 18.2% (50 PFU), or, 57.1% (10 PFU); and for ∆NSs-∆NSm-rZH501, 57.1% (1000 PFU), 14.3% (50 PFU), or 28.6% (10 PFU), respectively (Figure 1). Unexpectedly, the 1000 PFU dose led to higher survival rates in mice when compared to the 50 PFU dose after challenging them with rMP-12, RVax-1, or ∆NSs-∆NSm-rZH501. With a lack of dose dependency, we could not measure LD_50_ values.

Mother mice were also challenged with either rMP-12, RVax-1, or ∆NSs-∆NSm-rZH501 (1000, 50, or 10 PFU). Although all adult mice challenged with rMP-12 or RVax-1 (1000, 50, or 10 PFU), or ∆NSs-∆NSm-rZH501 (1000 or 10 PFU) survived without clinical signs of disease, an adult mouse challenged with 50 PFU of ∆NSs-∆NSm-rZH501 succumbed to infection at 8 dpc.

### 3.2. Viral Load of 19-Day-Old C57BL/6 Mice Succumbed to Live-Attenuated Rift Valley Fever Virus Infections

We assessed the viral loads in selected mice challenged with 50 PFU of viruses, which succumbed to the infections, namely rMP-12 (*n* = 3), RVax-1 (*n* = 4), and ∆NSs-∆NSm-rZH501 (*n* = 6). As presented in Table 2, abundant viral RNA (genomic M RNA) was detected in the brains of all examined mice. Although the timing of sample collections varied, the viral loads in the brain tissues of mice challenged with rMP-12, RVax-1, or ∆NSs-∆NSm-rZH501 ranged from 9.7 × 10^6^ to 2.4 × 10^9^, 7.9 × 10^7^ to 1.3 × 10^11^, and 1.2 × 10^8^ to 8.4 × 10^8^ copies/mg, respectively. In contrast, at least 10,000-fold less viral RNA were detected in the livers and spleens. A mouse infected with RVax-1 (mouse #2–3) exhibited relatively high viral loads not only in the brain but also in the liver and spleen tissues.

We also measured infectious virus titers in the livers, spleens, and brains of the infected mice. Despite a detection limit of 1.0 × 10^2^ PFU per 100 mg of tissue in our plaque assay, the livers of three RVax-1-infected mice showed detectable virus titers: 3.0 × 10^2^ PFU/100 mg (#2–2), 1.4 × 10^5^ PFU/100 mg (#2–3), and 4.0 × 10^2^ PFU/100 mg (#2–4). Similarly, the spleens of two RVax-1-infected mice had detectable virus titers: 3.4 × 10^4^ PFU/100 mg (#2–3) and 1.0 × 10^2^ PFU/100 mg (#2–4). No infectious viruses were detected in the livers or spleens of mice infected with either rMP-12 or ΔNSs-ΔNSm-rZH501. Meanwhile, the brains of mice challenged with rMP-12, RVax-1, or ∆NSs-∆NSm-rZH501 ranged from 2.2 × 10^6^ to 3.2 × 10^6^, 3.2 × 10^5^ to 2.9 × 10^8^, and 7.0 × 10^4^ to 1.7 × 10^5^ PFU/100 mg, respectively. This finding indicates that all three live-attenuated strains retain neurotropism in 19-day-old C57BL/6 mice. The results also showed that mice challenged with rMP-12 or RVax-1 exhibited varied level of viral loads in their brains in comparison to those challenged with ∆NSs-∆NSm-rZH501.

### 3.3. Histopathological Analysis of 19-Day-Old C57BL/6 Mice Succumbed to Live-Attenuated Rift Valley Fever Virus Infections

We conducted histological analysis on mouse tissues to characterize the pathological lesions linked with virus infections. Immunohistochemistry employing anti-RVFV N antibody, which can detect rMP-12, RVax-1, and ∆NSs-∆NSm-rZH501, revealed that virus infections in the brain of mice challenged with rMP-12, RVax-1, or ∆NSs-∆NSm-rZH501 were more widespread in the thalamus, hypothalamus, midbrain, and pons compared to the cortex, hippocampus, or cerebellum, whereas little antigen was detected in the livers and spleens of the challenged mice (Table 3).

Overall, the livers of mice infected with rMP-12, RVax-1, or ∆NSs-∆NSm-rZH501 did not show prominent pathological changes. To evaluate the differences among them, we stained liver sections with the anti-CD68 antibody, which is expressed in monocytes and macrophages, including Kupffer cells (Figure 2a–d). The results confirmed an increased number of CD68-positive cells in mice infected with RVax-1 (#2–2: 7.4-fold) and ∆NSs-∆NSm-rZH501 (#3–3: 9.5-fold, #3–4: 15.2-fold, and #3–5: 9.7-fold) (Figure 2e). Additionally, the relative sizes of CD68-positive signals were larger in mice infected with ∆NSs-∆NSm-rZH501 (#3–1: 3.0-fold) compared to those in mock-infected mouse.

The histopathological lesions in the brains of mice infected with rMP-12 and RVax-1 were similar (Figure 3a–d). There was minimal infiltration of inflammatory cells in the brain sections of mice infected with rMP-12 or RVax-1. However, viral antigens are abundantly detected in neurons and astrocytes, either focally or diffusely. Cytoplasmic vacuoles are occasionally present in neurons corresponding to the affected lesions. In contrast, the brain lesions of mice infected with ∆NSs-∆NSm-rZH501 were characterized by the infiltration and aggregation of neutrophils surrounding infected neurons (Figure 3e–h). Dead neutrophils exhibited pyknosis and karyorrhexis. Necrosis of neuropils was also observed in the affected areas (Figure 3e–h), whereas overall viral antigens were less abundant compared to those in mice infected with rMP-12 or RVax-1.

## 4. Discussion

In this study, we aimed to assess the relative virulence of three live-attenuated strains of the RVFV in highly susceptible 19-day-old mice. The rMP-12 strain is attenuated by several point mutations, including Y259H (Gn), R1182G (Gc), and R1029K (L), while the major virulence factor NSs gene remains functional [11]. The mechanism of MP-12 attenuation is not fully understood, but temperature sensitivity, which restricts replication above 38.5 °C, may contribute to the phenotype [12]. RVax-1 was generated from the rMP-12 strain and additionally includes (i) deletions of the 78 kD and NSm genes, and (ii) 566 silent mutations in the N, NSs, M, and L ORFs [13]. It was shown that the reassortant ZH501 strain encoding either the S-, M-, or L-segment of RVax-1 displayed a more attenuated phenotype compared to reassortant ZH501 strains that encoded the corresponding segments from MP-12 [11,14]. In contrast, the ∆NSs-∆NSm-rZH501 strain encodes a double deletion of the NSs and 78 kD/NSm genes, while the Gn/Gc and L genes remain wildtype. Given that the NSs gene is responsible for the viral evasion from host innate immune responses [34,35], the replication of an RVFV strain lacking the NSs gene induces robust type-I IFNs [18]. Additionally, the deletion of the 78 kD and NSm genes can diminish the hepatotropism of the RVFV in mice and rats [28,36].

Mice are highly susceptible to the RVFV infection and have been widely used as experimental models for studying viral pathogenesis and vaccinology. Wildtype RVFV strains are highly pathogenic to C57BL/6 mice, causing acute liver necrosis and late-onset viral encephalitis with 100% mortality at a dose as low as 1 PFU [24]. Due to their extreme susceptibility to the RVFV infection, we explored an approach to assess the relative virulence of live-attenuated RVFV vaccine candidates in mice. As reported, the MP-12 strain is highly attenuated in adult mice, with an LD_50_ of ≥10^6.3^ PFU in 4- to 6-week-old outbred ICR Swiss mice via i.p. inoculation [7]. Meanwhile, preweaning 19-day-old mice have been shown to be highly susceptible to the MP-12 strain, with an LD_50_ of 75.9 PFU in CD1 mice via i.p. inoculation [10]. In this study, we assessed preweaning 19-day-old inbred C57BL/6 mice, rather than outbred CD1 mice, to maintain a uniform genetic background for comparison among groups. We analyzed the infection of rMP-12, RVax-1, and ∆NSs-∆NSm-rZH501 in the preweaning mouse model to determine their relative overall virulence (e.g., LD_50_) and the unique pathological changes in affected organs. Three different doses were used to evaluate the dose-dependent virulence of rMP-12, RVax-1, and ∆NSs-∆NSm-rZH501: i.e., 1000, 50, and 10 PFU. As anticipated, preweaning 19-day-old C57BL/6 mice exhibited significant susceptibility to rMP-12, RVax-1, and ∆NSs-∆NSm-rZH501. Notably, the severity of the illness did not correlate with viral doses; specifically, administering a 1000 PFU dose resulted in higher survival rates among mice compared to a 50 PFU dose. Haddock et al. [37] reported that C57BL/6CNCrl and C57BL/6J mice infected with a high dose (1000 focus-forming units, FFU) of mouse-adapted Ebolavirus (MA-EBOV) exhibited increased survival compared to those infected with a lower dose (10 FFU). However, this increased survival at 1000 FFU was not observed in C57BL/6N or CD1 outbred mice infected with MA-EBOV. Although the underlying mechanisms related to mouse strain differences have not been fully characterized, it is suggested that a higher viral load may likely lead to a more rapid and robust activation of innate immune responses [37]. Dodd et al. demonstrated that in 8- to 10-week-old C57BL/6 mice, the clearance of ∆NSs-rZH501 is primarily mediated by humoral immunity via CD4+ T cells, rather than cellular immunity by CD8+ T cells [38]. Therefore, the increased survival observed with 1000 PFU may also be associated with early humoral immune responses in the RVFV infection. We additionally administered a dose of 1 PFU of rMP-12 (*n* = 7, back-titration titer: 0.5 PFU) and RVax-1 (*n* = 7, back-titration titer: 1.0 PFU) to 19-day-old mice, resulting in the survival of 4 and 6 mice, respectively (Appendix A). There was a clear dose dependency observed among the 1, 10, and 50 PFU doses. Excluding the 1000 PFU dose, the LD50 values for rMP-12, RVax-1, and ∆NSs-∆NSm-rZH501 were calculated to be 16.0, 13.7, and less than 7.0, respectively [39]. However, it would be beneficial to test other inbred mouse models to enhance the accuracy and range of LD_50_ measurements.

It is noteworthy that mice infected with ∆NSs-∆NSm-rZH501 exhibited lower body weight gains in comparison to those infected with rMP-12 or RVax-1 and succumbed to infection between 6 and 8 dpc. Meanwhile, mice infected with rMP-12 or RVax-1 steadily gained body weight prior to the onset of neurological signs. Since both RVax-1 and ∆NSs-∆NSm-rZH501 lack 78 kD and NSm genes, this difference might be ascribed to the NSs protein’s function. Robust viral replication in the absence of the NSs protein can provoke host innate immune responses, such as type I IFNs, whereas the replication of rMP-12 or RVax-1 occurs with minimal stimulation of host innate responses. Histopathological findings corroborated the clinical observations. Infections with ∆NSs-∆NSm-rZH501 induced innate immune responses, activating macrophages and neutrophils. Despite the lower abundance of ∆NSs-∆NSm-rZH501 antigens compared to rMP-12 or RVax-1 in the brains, active neutrophil infiltration and apoptosis resulted in focal necrosis of neuropils in the midbrain or hippocampus. The presence of neutrophils in brain parenchyma of mice infected with ∆NSs-rZH501 has been reported in past studies by others [40]. In contrast, infection with rMP-12 or RVax-1 resulted in a more widespread viral spread within neurons and astrocytes, without active neutrophil infiltration or significant neuropil necrosis. Infected neurons exhibited vacuolar changes in the cytoplasm, which are potentially linked to viral neuropathy induced by these strains in preweaning C57BL/6 mice. Meanwhile, no lesions were detected in the livers; however, there was an increase in CD68-positive cells in some mice infected with ∆NSs-∆NSm-rZH501, compared to those in the mock-infected livers. CD68 is highly expressed in monocytes and macrophages, including Kupffer cells. Although the pathological role of CD68-positive cells in livers remains unknown in our study, further characterization of these cells may enhance our understanding of the host antiviral response in livers during RVFV infection.

The RVFV displays diverse host tropism: (i) highly susceptible hosts include mice, hamsters, newborn lambs, and goat kids; (ii) moderately susceptible hosts encompass humans, nonhuman primates, sheep, cattle, and goats; (iii) minimally susceptible hosts include horses, pigs, and rabbits; and (iv) refractory hosts include guinea pigs. Notably, preweaning mice are even more susceptible than adult mice, highlighting the importance of a careful interpretation of this study’s results. This increased susceptibility makes them particularly useful for evaluating live-attenuated RVFV strains, rather than wildtype pathogenic strains. The preweaning mouse model can be valuable for assessing the residual virulence or potential reversion to virulence of live-attenuated vaccine strains, including different vaccine lots and vaccine strain isolates. However, the high fatality rate in preweaning mice is likely indicative of viral neuroinvasiveness and viral encephalitis. The distinct pathogenesis observed between rMP-12 (or RVax-1) and ∆NSs-∆NSm-rZH501 in this study will inform the future characterization of similar live-attenuated RVF vaccines. Given that this model exhibited a lack of dose-dependent virulence, further refinement may be required, such as optimizing mouse strains or adjusting the route or timing of virus challenge. Nevertheless, the preweaning mouse model is likely to be valuable for evaluating the host antiviral response patterns to attenuated RVFV strains.

## Figures and Tables

**Figure 1 viruses-16-00999-f001:**
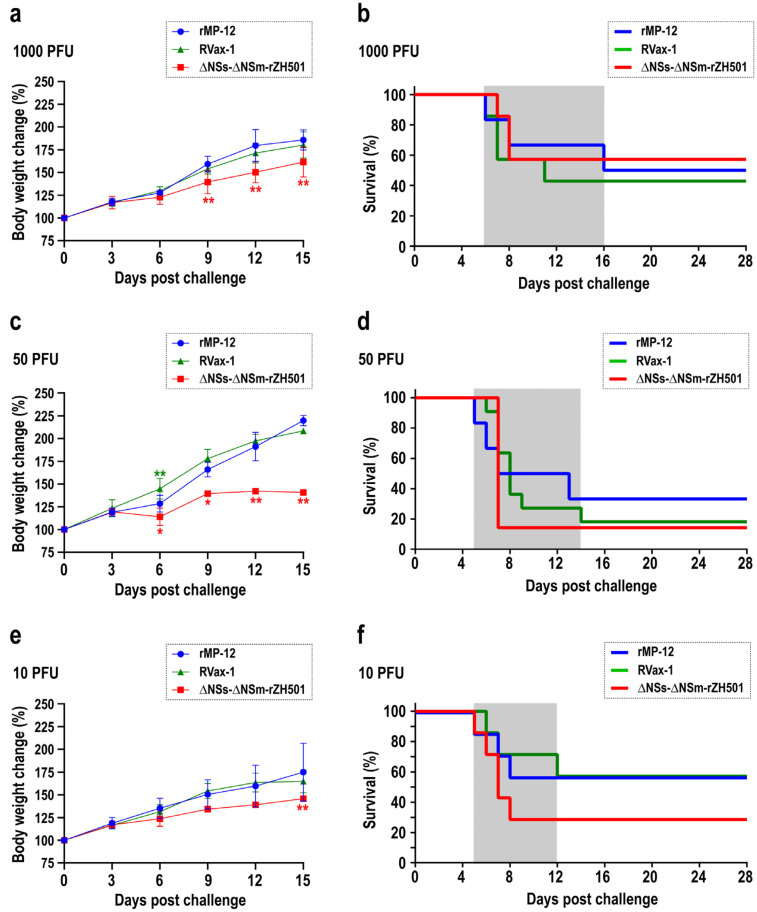
The susceptibility of preweaning mice to three strains of Rift Valley fever virus: rMP-12, RVax-1, or ∆NSs-∆NSm-rZH501. The graph displays changes in body weight (%) (**a**,**c**,**e**) and Kaplan–Meier survival curves (**b**,**d**,**f**) of 19-day-old C57BL/6 mice challenged intraperitoneally (i.p.) with specified doses of rMP-12, RVax-1, or ∆NSs-∆NSm-rZH501. Mice were administered either 1000 PFU (**a**,**b**), 50 PFU (**c**,**d**), or 10 PFU (**e**,**f**) of rMP-12 (blue), RVax-1 (green), or ∆NSs-∆NSm-rZH501 (red). Mice exhibiting over 20% body weight loss and/or severe clinical symptoms were euthanized, while surviving mice were euthanized at 28 days post-challenge (dpc). Error bars represent the standard deviations. Asterisks denote statistical significance compared to the rMP-12 group (* *p* < 0.05, ** *p* < 0.01, two-way ANOVA).

**Figure 2 viruses-16-00999-f002:**
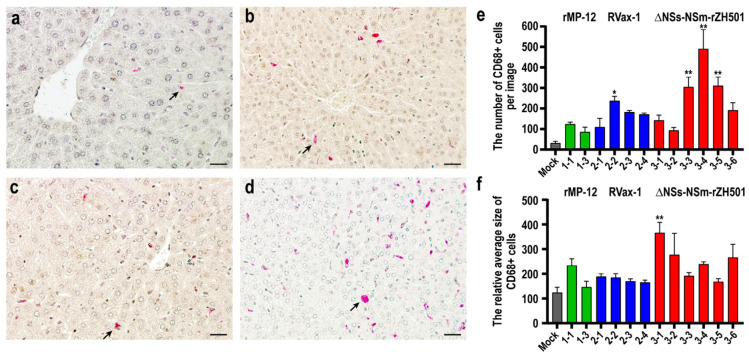
CD68-positive cells in livers of preweaning mice infected with Rift Valley fever virus: rMP-12, RVax-1, or ∆NSs-∆NSm-rZH501. Liver sections from C57BL/6 mice were either mock-infected (**a**) or infected with rMP-12 ((**b**): animal #1–1), RVax-1 ((**c**): animal #2–1), or ∆NSs-∆NSm-rZH501 ((**d**): animal #3–1). These sections were stained with anti-CD68 antibody via immunohistochemistry. The scale bar represents 50 µm, and black arrows indicate CD68-positive cells. (**e**) The graph shows the mean numbers with standard errors of CD68-positive cells in four different liver images at ×20 magnification. (**f**) The graph displays the relative average sizes with standard errors of CD68-positive signals used for the cell count in graph (**e**). An asterisk represents statistical significance determined by one-way ANOVA; * *p* < 0.05 or ** *p* < 0.01, compared to the mock-infected group.

**Figure 3 viruses-16-00999-f003:**
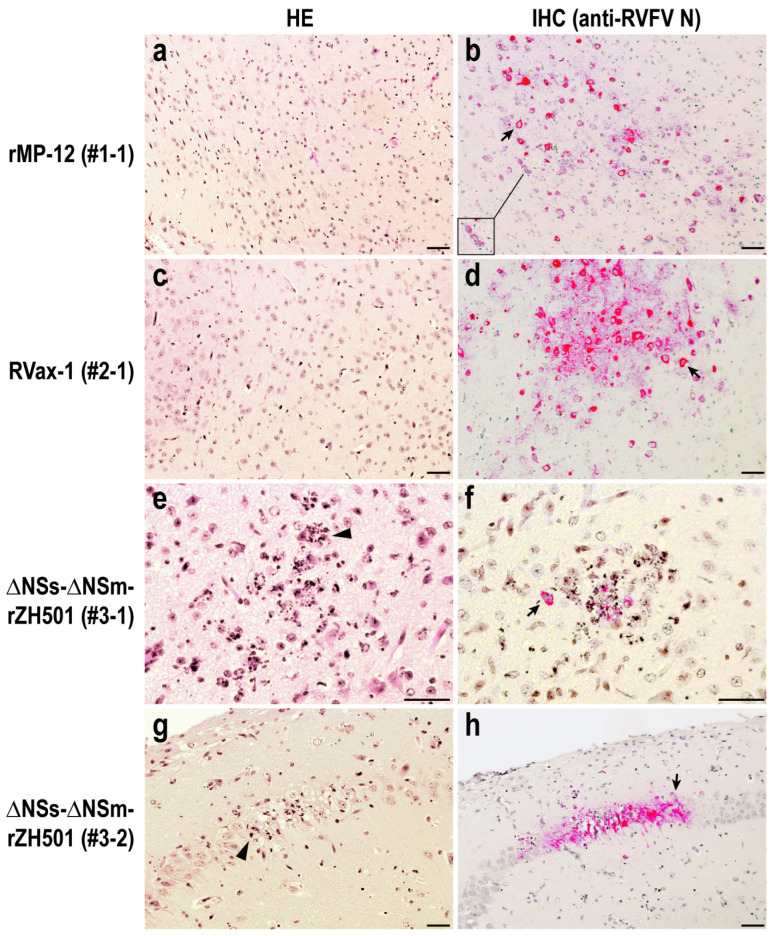
Brain lesions in preweaning mice infected with Rift Valley fever virus: rMP-12, RVax-1, or ∆NSs-∆NSm-rZH501. Sections of midbrain (**a**–**f**) and hippocampus (**g**,**h**) were stained with hematoxylin and eosin (HE: (**a**,**c**,**e**,**g**)) or anti-RVFV N antibody by immunohistochemistry (IHC: (**b**,**d**,**f**,**h**)). The animal ID# and viruses are indicated on the left. Scale bar represents 50 µm. An inset (**b**) shows cytoplasmic vacuoles in infected neurons. Black arrows indicate RVFV N-positive cells (stained in red); arrowheads indicate infiltrations and aggregations of neutrophils with pyknosis and karyorrhexis.

**Table 1 viruses-16-00999-t001:** Experimental groups of C57BL/6 mice: sex, viral strains, target and actual doses.

Group	Sex (*n* ^1^)	Viruses	Target Doses (PFU ^2^)	Actual Doses (PFU)
1	M (2)/F (4)	rMP-12	1000	1680
2	M (2)/F (4)	rMP-12	50	84
3	M (3)/F (4)	rMP-12	10	8
4	M (2)/F (5)	RVax-1	1000	840
5	M (4)/F (7)	RVax-1	50	36
6	M (3)/F (4)	RVax-1	10	9
7	M (5)/F (2)	∆NSs-∆NSm-rZH501	1000	567
8	M (3)/F (4)	∆NSs-∆NSm-rZH501	50	52
9	M (4)/F (3)	∆NSs-∆NSm-rZH501	10	7

^1^ *n*: the number of 19-day-old mice, ^2^ PFU: Plaque-forming unit.

**Table 2 viruses-16-00999-t002:** Viral loads of preweaning mice succumbed to live-attenuated RVFV infections.

Viruses	Animal ID# (PFU Dose ^1^; Death ^2^)	Liver (Copies/mg) ^3^	Spleen (Copies/mg)	Brain (Copies/mg)
		[PFU/100 mg] ^4^	[PFU/100 mg]	[PFU/100 mg]
rMP-12	1–1 (50; 5 dpc)	5.0 × 10^1^	N.T. ^5^	9.7 × 10^6^
		[<1.0 × 10^2^]		[2.2 × 10^6^]
	1–2 (50; 6 dpc)	7.5 × 10^1^	N.T.	2.4 × 10^9^
	1–3 (50; 13 dpc)	4.5 × 10^1^	1.1 × 10^1^	1.0 × 10^9^
		[<1.0 × 10^2^]	[<1.0 × 10^2^]	[3.2 × 10^6^]
RVax-1	2–1 (50; 7 dpc)	2.4 × 10^3^	N.T.	1.8 × 10^8^
		[<1.0 × 10^1^]	N.T.	[1.4 × 10^6^]
	2–2 (50; 7 dpc)	2.5 × 10^2^	N.T.	7.9 × 10^7^
		[3.0 × 10^2^]	N.T.	[3.2 × 10^5^]
	2–3 (50; 8 dpc)	3.6 × 10^5^	6.9 × 10^5^	1.3 × 10^11^
		[1.4 × 10^5^]	[3.4 × 10^4^]	[2.9 × 10^8^]
	2–4 (50; 8 dpc)	4.3 × 10^4^	2.9 × 10^3^	9.6 × 10^8^
		[4.0 × 10^2^]	[1.0 × 10^2^]	[9.7 × 10^6^]
∆NSs-∆NSm	3–1 (50; 7 dpc)	6.1 × 10^2^	3.1 × 10^2^	5.4 × 10^8^
-rZH501		[<1.0 × 10^2^]	[<1.0 × 10^2^]	[1.2 × 10^5^]
	3–2 (50; 7 dpc)	6.1 × 10^2^	8.7 × 10^1^	7.7 × 10^8^
		[<1.0 × 10^2^]	[<1.0 × 10^2^]	[9.0 × 10^4^]
	3–3 (50; 7 dpc)	9.8 × 10^0^	4.5 × 10^2^	2.5 × 10^8^
		[<1.0 × 10^2^]	[<1.0 × 10^2^]	[7.0 × 10^4^]
	3–4 (50; 7 dpc)	1.5 × 10^2^	2.0 × 10^2^	8.4 × 10^8^
		[<1.0 × 10^2^]	[<1.0 × 10^2^]	[8.0 × 10^4^]
	3–5 (50; 7 dpc)	3.3 × 10^2^	2.5 × 10^2^	7.6 × 10^8^
		[<1.0 × 10^2^]	[<1.0 × 10^2^]	[1.7 × 10^5^]
	3–6 (50; 7 dpc)	2.8 × 10^2^	1.1 × 10^2^	1.2 × 10^8^
		[<1.0 × 10^2^]	[<1.0 × 10^2^]	[1.0 × 10^5^]

^1^ Virus dose, plaque-forming unit (PFU) per mL, upon the virus challenge; ^2^ the day of death post-challenge; ^3^ RVFV M-segment RNA copy number per mg tissues, determined by the RT-qPCR; ^4^ infectious virus titer, PFU per mL of 10% tissue homogenates with 100 mg tissue; ^5^ not tested.

**Table 3 viruses-16-00999-t003:** Viral nucleocapside antigen distributions of preweaning mice challenged with live-attenuated RVFV.

Viruses 50 PFU Dose	Animal ID# (Death)	Liv	Spl	Cor	Hp	Tha	Mid	Pon	Cer
rMP-12	1–1 (5 dpc)	−	−	−	−	++	+++	+++	N.T.
	1–3 (13 dpc)	−	−	−	−	+	+	N.T.	N.T.
RVax-1	2–1 (7 dpc)	−	−	−	−	+	+++	+++	+
	2–2 (7 dpc)	−	^−^	−	−	+	+	++	++
	2–3 (8 dpc)	+	−	+++	+++	+++	N.T.	N.T.	N.T.
	2–4 (8 dpc)	−	−	−	−	++	+	N.T.	N.T.
∆NSs-∆NSm	3–1 (7 dpc)	−	−	−	−	++	+	+++	++
-rZH501	3–2 (7 dpc)	−	−	+	++	++	+	++	+
	3–3 (7 dpc)	−	−	+	−	+	++	+++	−
	3–4 (7 dpc)	−	−	+	−	++	++	++	+
	3–5 (7 dpc)	−	−	−	−	−	+	+	−

+++, diffuse; ++, focal; +, isolated; −, not detectable; N.T., not tested. Liv, liver; Spl, spleen; Cor, cortex; Hp, hippocampus; Tha, thalamus/hypothalamus; Mid, midbrain; Pon, pons; Cer, cerebellum.

## Data Availability

Data and materials are available through the agreement term made via the Office of Technology Transfer at UTMB.

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
