# Peer review of "Distinct Pathological Changes in Preweaning Mice Infected with Live-Attenuated Rift Valley Fever Virus Strains"

_viruses, 2024, doi:10.3390/v16070999_

Round 1

Reviewer 1 Report

Comments and Suggestions for Authors

In this manuscript, Cigdem Alkan et al. compared three live-attenuated RVF vaccine strains, namely, rMP-12, RVax-1, and the NSs-NSm-rZH501 in the 19-day-old inbred C57BL/6 mice. Which added new information about the virulence of these live-attenuated RVF vaccine strains.

The overall result seems sound and supported by the results presented. I have no other questions but am curious about the peculiar result obtained from the highest dose of RVFV (1000 PFUs) , which results in a higher survival rate compared to the 50 PFUs dose. And the author discussed that “An increased survival of mice with 1,000 PFU suggests that host immunity may impede viral evasion into the central nervous system (CNS). ”

To better illustrate this result, I would suggest the author to briefly summarize the previous studies that used different doses and related outcomes either in inbred or outbred mice models, which may contribute to more possibilities that could address this conclusion.

Author Response

In this manuscript, Cigdem Alkan et al. compared three live-attenuated RVF vaccine strains, namely, rMP-12, RVax-1, and the NSs-NSm-rZH501 in the 19-day-old inbred C57BL/6 mice. Which added new information about the virulence of these live-attenuated RVF vaccine strains.

The overall result seems sound and supported by the results presented. I have no other questions but am curious about the peculiar result obtained from the highest dose of RVFV (1000 PFUs) , which results in a higher survival rate compared to the 50 PFUs dose. And the author discussed that “An increased survival of mice with 1,000 PFU suggests that host immunity may impede viral evasion into the central nervous system (CNS). ”

To better illustrate this result, I would suggest the author to briefly summarize the previous studies that used different doses and related outcomes either in inbred or outbred mice models, which may contribute to more possibilities that could address this conclusion.

[Response] Thank you for your comments. We have now cited a relevant publication that describes the varying survival rates between inbred and outbred mice infected with mouse-adapted Ebolavirus (Haddock E. et al.) in Discussion.

Thank you for your review and suggestions.

Reviewer 2 Report

Comments and Suggestions for Authors

This interesting study aimed to characterise the virulence of three RVF atenuatted vaccine candidates. The authors used a 19-day-old preweaning mouse model because of its high susceptibility to RVFV infection, which allows the virulence of attenuated vaccines such as those described in the study to be assessed. Although the observed differences are indeed subtle, they are sufficiently evident to suggest this model as a useful tool for virulence assessment. Other authors have shown that mice deficient in type I interferon signalling pathways may also be suitable for similar purposes. One of the conclusions of the paper is that the function of the NSs protein seems essential to allow some virus replication without provoking an excessive inflammatory response. A fine balance between replication and infammation determines the greater or lesser degree of virulence in these models. In all cases the three viruses retained neurovirulence although with distinctive features.

Specific comments

On page 7 lines 231. Explain better the meaning of this sentence. The table 2 show differences in PCR virus loads between rMP12 and Rvax-1 in liver and spleen. In fact, they seem more evident than for the double deleted virus.

The authors demonstrate the presence of viral genome by RT-qPCR in tissues. It would be very interesting to know whether it was attempted to isolate infectious virus from these tissues. Alternatively, explain how  the detected qPCR levels correlate with  the presence of infectious virus.

It would also be very  interesting to know whether viruses isolated from brain tissues maintain the original sequence or whether or not have significant sequence changes.

It is claimed that viruses devoid of NSm expression are less hepatotropic. This would explain why less viral antigen is detected in livers from both delta NSm virus infected mice. Another possible explanation could be that the times post infection tested were already late for detection in this organ?

The fact that a higher dose of inoculation is less lethal has been reported previously by other studies in mice. A possible explanation for this phenomenon could be the presence in the inoculum of more defective particles capable of inducing a stronger host immunity,  restricting virus replication.

Minor comments

Congratulate the authors for showing in table 1 the results of the actual doses inoculated to the mice.

Line 208, on page 5, should use “With” instead of “Without”

In figure 1 placing the pfu doses to the left of the graphs would improve the presentation.

In table 2, the single asterisk refers to the weight of the mouse, but it actually refers to the dose.

On line 247 on page 7 it should read “little antigen” instead of “little antigens”

Author Response

This interesting study aimed to characterise the virulence of three RVF atenuatted vaccine candidates. The authors used a 19-day-old preweaning mouse model because of its high susceptibility to RVFV infection, which allows the virulence of attenuated vaccines such as those described in the study to be assessed. Although the observed differences are indeed subtle, they are sufficiently evident to suggest this model as a useful tool for virulence assessment. Other authors have shown that mice deficient in type I interferon signalling pathways may also be suitable for similar purposes. One of the conclusions of the paper is that the function of the NSs protein seems essential to allow some virus replication without provoking an excessive inflammatory response. A fine balance between replication and infammation determines the greater or lesser degree of virulence in these models. In all cases the three viruses retained neurovirulence although with distinctive features.

Specific comments

On page 7 lines 231. Explain better the meaning of this sentence. The table 2 show differences in PCR virus loads between rMP12 and Rvax-1 in liver and spleen. In fact, they seem more evident than for the double deleted virus.

[Response] We revised the section 3.2. in the revised manuscript to clarify the difference between the rMP-12 and RVax-1 results.

The authors demonstrate the presence of viral genome by RT-qPCR in tissues. It would be very interesting to know whether it was attempted to isolate infectious virus from these tissues. Alternatively, explain how  the detected qPCR levels correlate with  the presence of infectious virus.

[Response] Infectious virus titers are newly added to Table 2.

It would also be very  interesting to know whether viruses isolated from brain tissues maintain the original sequence or whether or not have significant sequence changes.

[Response] Thank you for suggesting potential changes in the viral genetic sequence. While we are also interested in exploring genetic changes through brain invasion, this aspect falls beyond the current scope of our study.

It is claimed that viruses devoid of NSm expression are less hepatotropic. This would explain why less viral antigen is detected in livers from both delta NSm virus infected mice. Another possible explanation could be that the times post infection tested were already late for detection in this organ?

[Response] Thank you for your comment on the 78kD/NSm deletion mutant phenotype. A previous study (Nishiyama et al., 2016) demonstrated that mice infected with rZH501 lacking the 78kD/NSm protein exhibited lower ALT levels in the blood at 3 dpi, along with reduced viral loads in the livers, compared to parental rZH501 strain.  

The fact that a higher dose of inoculation is less lethal has been reported previously by other studies in mice. A possible explanation for this phenomenon could be the presence in the inoculum of more defective particles capable of inducing a stronger host immunity,  restricting virus replication.

 [Response] Thank you for the comment. This point was also raised by Reviewer #1. We have cited a relevant publication that describes the varying survival rates between inbred and outbred mice infected with mouse-adapted Ebolavirus (Haddock E. et al.) in Discussion.

Minor comments

Congratulate the authors for showing in table 1 the results of the actual doses inoculated to the mice.

Line 208, on page 5, should use “With” instead of “Without”

 [Response] Corrected.

In figure 1 placing the pfu doses to the left of the graphs would improve the presentation.

[Response] Corrected as suggested.

In table 2, the single asterisk refers to the weight of the mouse, but it actually refers to the dose.

[Response] Corrected.

On line 247 on page 7 it should read “little antigen” instead of “little antigens”

[Response] Corrected as suggested.

Thank you for your review and suggestions.

Reviewer 3 Report

Comments and Suggestions for Authors

Author Response

In this work, authors carried out an experimental infection in 19-day-old preweaning inbred C57BL/6 mice with three different RVFV live-attenuated vaccines using three different doses 1000, 500 and 10 pfu to characterize its susceptibility and virulence. They analyzed the change in body weight, the survival curve and viral RNA loads in different tissues during 28 days postchallenge. Results showed that preweaning inbred mice exhibited high susceptibility to all three strains with a higher survival rates observed in mice with the 1,000 PFU dose compared with the other doses. Also they observed that all three live-attenuated strains retain neurotropism in 19-day-old C57BL/6 mice. While the authors have provided detailed results in their manuscript, there are several issues that need to be addressed for clarity and qualification of the study's impact and significance. The manuscript would benefit from revision to rectify these concerns. Please, respond to the next questions:

Comments:

Authors could answer why they chose these three doses of virus.

[Response] We have revised the second paragraph of the Discussion to provide a more detailed explanation of dose dependency and its interpretation. Thank you for your comments.

Authors could explain why the most attenuated vaccine NSs-NSm-rZH501 shows survival rates lower than the other strains and pathological changes in organs with the three doses assayed.

[Response] We revised the Discussion (3rd paragraph) to explain our thought regarding the pathology of NSs-NSm-rZH501 infection.

In the table 1, authors showed the back-titration obtained by plaque assay with the 1000 pfu inoculum is the half of the initial dose. Have the authors corrected the results obtained with this mice group?

Addressing these critical issues would significantly enhance the quality, clarity, and impact of the manuscript.

[Response] Thank you for your suggestion. Following the back-titration, we tested 1 PFU of rMP-12 and RVax-1 to explore dose-dependent responses, rather than repeating 1,000 PFU. This information has been incorporated into the second paragraph of the Discussion section.

Thank you for your review and suggestions.